# Bispecific Antibodies in Hematological Malignancies: A Scoping Review

**DOI:** 10.3390/cancers15184550

**Published:** 2023-09-14

**Authors:** Mohamed H. Omer, Areez Shafqat, Omar Ahmad, Khaled Alkattan, Ahmed Yaqinuddin, Moussab Damlaj

**Affiliations:** 1School of Medicine, Cardiff University, Cardiff CF14 4YS, UK; 2College of Medicine, Alfaisal University, Riyadh 11533, Saudi Arabia; ashafqat@alfaisal.edu (A.S.); oahmad@alfaisal.edu (O.A.); kkattan@alfaisal.edu (K.A.); ayaqinuddin@alfaisal.edu (A.Y.); 3Department of Hematology & Oncology, Sheikh Shakhbout Medical City, Abu Dhabi P.O. Box 11001, United Arab Emirates; mdamlaj@gmail.com; 4College of Medicine, Khalifa University, Abu Dhabi P.O. Box 127788, United Arab Emirates

**Keywords:** bispecific antibody, antibodies, CAR-T, lymphoma, leukemia, multiple myeloma, hematological cancer

## Abstract

**Simple Summary:**

Bispecific T-cell engagers (BiTEs) and bispecific antibodies (BiAbs) have emerged as novel therapeutic modalities in the treatment of advanced hematological malignancies. BiTEs and BiAbs redirect T cells to attack tumors and facilitate T-cell-mediated cell death. Blinatumomab was the first BiTE to display proof-of-concept with its remarkable contribution towards the treatment of acute lymphoblastic leukemia. Nearly a decade later, several BiTEs/BiAbs targeting a range of tumor-associated antigens have transpired in the treatment of multiple myeloma, non-Hodgkin’s lymphoma, acute myelogenous leukemia, and acute lymphoblastic leukemia. This review summarizes the most recent evidence emerging from clinical trials regarding the use of BiAbs and BiTEs in hematological malignancies whilst highlighting the limitations of these therapeutic options and providing practical insights towards overcoming these limitations.

**Abstract:**

Bispecific T-cell engagers (BiTEs) and bispecific antibodies (BiAbs) have revolutionized the treatment landscape of hematological malignancies. By directing T cells towards specific tumor antigens, BiTEs and BiAbs facilitate the T-cell-mediated lysis of neoplastic cells. The success of blinatumomab, a CD19xCD3 BiTE, in acute lymphoblastic leukemia spearheaded the expansive development of BiTEs/BiAbs in the context of hematological neoplasms. Nearly a decade later, numerous BiTEs/BiAbs targeting a range of tumor-associated antigens have transpired in the treatment of multiple myeloma, non-Hodgkin’s lymphoma, acute myelogenous leukemia, and acute lymphoblastic leukemia. However, despite their generally favorable safety profiles, particular toxicities such as infections, cytokine release syndrome, myelosuppression, and neurotoxicity after BiAb/BiTE therapy raise valid concerns. Moreover, target antigen loss and the immunosuppressive microenvironment of hematological neoplasms facilitate resistance towards BiTEs/BiAbs. This review aims to highlight the most recent evidence from clinical trials evaluating the safety and efficacy of BiAbs/BiTEs. Additionally, the review will provide mechanistic insights into the limitations of BiAbs whilst outlining practical applications and strategies to overcome these limitations.

## 1. Introduction

T-cell-redirecting strategies have emerged as highly promising therapeutic modalities for the treatment of hematological malignancies. Notably, two approaches, namely chimeric antigen receptor T cells (CAR-T) and bispecific antibodies (BiAbs), have shown remarkable efficacy in the treatment of hematological malignancies. CAR-T cells have revolutionized the management of relapsed and refractory hematological malignancies like multiple myeloma (MM), non-Hodgkin’s lymphoma (NHL), and acute lymphoblastic leukemia (ALL) [1,2,3,4]. Despite CAR-T therapy’s remarkable success, its lengthy engineering process—spanning approximately 6–8 weeks—can render some patients with advanced disease ineligible for this therapy [5]. Furthermore, CAR-T therapy is often associated with multiple end-organ toxicities, including severe neurotoxicity and cytokine release syndrome (CRS), which may limit its utility, especially in patients with a lower performance status or other comorbidities [6].

On the other hand, BiAbs and bispecific T-cell engagers (BiTEs) offer the T-cell-redirecting capabilities of CAR-T as an off-shelf therapy whilst eliminating the logistical and time constraints associated with CAR-T delivery. Additionally, BiAbs and BiTEs appear to have a more favorable safety profile, with lower incidences of CRS and neurotoxicity than CAR-T therapy [7].

The pioneering success of blinatumomab as the first BiTE demonstrated proof-of-concept evidence with its remarkable contribution towards the treatment of ALL [8]. Nearly a decade later, several BiAbs and BiTEs have emerged in the therapeutic landscape of hematological neoplasms. This review summarizes the latest clinical trial evidence regarding the use of BiAbs and BiTEs in hematological malignancies. Furthermore, it aims to highlight the limitations associated with these therapeutic options and provide practical insights towards overcoming these limitations.

## 2. The Biology of Bispecific Antibodies

Given that our focus is summarizing clinical data on the use of BiAbs in hematological malignancies, a detailed discussion of the basic science of these drugs is beyond the scope of this review and we direct readers towards other reviews that have explored this area in greater detail [9,10].

### 2.1. Mechanism of Action

CAR-T therapy involves engineering a T cell to express a chimeric antigen receptor (CAR) specific to a tumor-associated antigen. Administering CAR-T cells would hence augment the anti-tumor immune response. BiAbs achieve a similar goal by containing two binding sites, enabling them to bind two epitopes on the same antigen or two different antigens [10,11]. One arm of the BiAb binds to the target tumor-associated antigen and the other simultaneously binds CD3 on the surfaces of CD4+ helper T cells and CD8+ cytotoxic T cells, resulting in the formation of an immunological synapse that activates T cells without the need for T cell recognition of the MHC/antigen complex on tumor cells [10]. Activated T cells release perforin and granzyme, resulting in the T-cell-dependent killing of tumor cells via apoptosis. While BiAbs are a broad category of antibodies that target two antigens or epitopes, the specific class of BiAbs that form immunological synapses between T cells and tumor cells are called bispecific T-cell engagers (BiTEs). The BiAbs discussed in this review are mostly BiTEs, and hence the two terms will be used interchangeably.

### 2.2. Resistance Mechanisms

Like other modalities of cancer treatment, tumors can become resistant to BiAbs/BiTEs and consequently impair therapeutic efficacy. The administration of BiAbs imposes significant selection pressures on tumor clones expressing the target tumor-associated antigen, but inadvertently confer a selective advantage to sub-clones lacking the target antigen, resulting in their expansion and resistance [12]. To counter this, combinatorial strategies of administering multiple BiAbs or trispecific antibodies that target an additional tumor antigen have been explored [12,13,14]. Additionally, specific genetic abnormalities in AML and ALL have been associated with an inferior response to BiAbs/BiTEs, but the underlying mechanism by which an adverse cytogenetic profile modulates the therapy response is unclear [15,16,17]. Specific tumor cells may also alter their intracellular signaling pathways in response to T-cell-redirecting therapy, as one study showed that disrupted interferon-gamma signaling in HER2-positive tumor cells conferred resistance to killing by BiTE/CAR-redirected T cells [18]. Alternatively, features extrinsic to the tumor cell, such as the strong presence of regulatory T cells in the tumor microenvironment (TME), have been shown to modulate the therapeutic response to BiAbs/BiTEs in multiple myeloma and B-cell ALL [19,20,21,22]. Lastly, resistance may develop as a consequence of prior lines of cancer treatment itself, which can lead to a loss of T cell fitness and anti-tumor function [23]. Long-term administration of BiTEs can continually stimulate T cells, promoting their exhaustion and thereby promoting tumor survival [24,25]. The specific mechanisms at play in different types of hematological malignancies are discussed in their respective sections below.

## 3. Bispecific T-Cell Engagers and Antibodies in the Treatment of Multiple Myeloma

The introduction of protease inhibitors, immunomodulatory drugs (IMIDs), and anti-CD38 monoclonal antibodies has significantly improved multiple myeloma (MM) patient outcomes [26], but those with high-risk disease and adverse cytogenetic profiles often do not respond to these treatments [27]. Such patients, often termed ‘triple refractory’, exhibit poor survival outcomes [28]. In this regard, BiAbs and BiTEs have emerged as promising additions to the MM treatment landscape, particularly for triple-refractory MM.

A prime target for BiTE therapy in MM is the B-cell maturation antigen (BCMA). BCMA is selectively expressed on the surfaces of plasma cells and is associated with disease severity and unfavorable prognostic outcomes [29,30]. Teclistamab, a humanized IgG BCMA-targeting BiAb, was recently approved by the European Medicines Agency (EMA) and the Food and Drug Administration (FDA) for relapsed-refractory multiple myeloma (RRMM) [31]. The MajesTEC-1 clinical trial demonstrated that teclistamab had an overall response rate (ORR) of 63.0% and a complete response in 39.4% of 165 patients during an average follow-up of 14.1 months [32]. The median duration of response and progression-free survival was 18.4 months and 11.3 months, respectively. Adverse events included grade 1–2 CRS, cytopenias, and infections [32]. Elranatamab, another BCMA targeting IgG2A BiAb, has also shown promise in RRMM patients. It has obtained orphan drug designation by the EMA and FDA. The MagnetisMM-3 trial demonstrated an ORR of 61.0%, with a complete response achieved in 27.6% of 123 enrolled patients with triple-refractory MM during a median follow-up duration of 6.8 months [33]. CRS was the most common side effect, with an otherwise manageable safety profile [33]. The recent phase 2 LINKER-MM1 clinical trial explored the use of the anti-BCMA BiTE linvoseltamab in triple-refractory MM, reporting an ORR of 64% in patients receiving a higher dose (200 mg) compared to 50% for those on the lower dose (50 mg). Linvoseltamab demonstrated a tolerable safety profile, with CRS and infections as the most common adverse events [34]. ABBV-383, an anti-BCMA BiAb/BiTE, has the advantage of not requiring step-up dosing, making it easier to administer and monitor. In a phase 1 trial, ABBV-383 achieved an ORR of 57% and a complete response in 29% of 124 RRMM patients [35].

The use of BCMA-targeting therapeutic modalities in MM may result in either the decreased or complete loss of BCMA expression on MM cells and consequent antigen escape [36]. To address this issue, researchers have explored the targeting of additional antigens with BiTEs. G-protein-coupled receptor family C group 5 member D (GPRC5D), expressed on neoplastic MM cells, is one such target [37]. The MonumenTAL-1 phase 1 clinical trial reported that talquetamab, an IgG4 Fc BiAb directed against GPRC5D, demonstrated an ORR of 64–70% in 232 heavily pretreated RRMM patients [38]. The main adverse events associated with talquetamab were CRS in 77–80% of patients (primarily grades 1–2) and hematologic toxicity [38]. Talquetamab was additionally associated with unique adverse effects of skin and nail disorders, likely explained by the expression of GPRC5D in keratinized tissues and hair follicles [38]. However, the majority of these particular adverse events were well tolerated in the MonumenTAL-1 trial [37]. Another GPRC5D targeting BiTE, RG6234, demonstrated an ORR of 71.4% in RRMM patients during a phase 1 clinical trial [39]. This trial also evaluated the response in patients who received prior BCMA-targeting BiAb therapy, demonstrating an ORR of 55.6% in these patients [39].

The Fc Receptor Homolog 5 (FcRH5) and CD38 have been considered potential targets for BiAbs in MM [40]. Cevostamab, a humanized anti-FcRH5 IgG1 BiAb, elicited treatment responses in patients previously exposed to CAR-T (44.4%), BiAbs (33.3%), antibody–drug conjugates (50.0%), and BCMA-targeted therapy (36.4%) in an ongoing phase 1 study enrolling 160 RRMM patients with a manageable safety profile, with grade 1–2 CRS being the most common side effect, indicating its potential as a salvage therapy [41]. CD38, a transmembrane glycoprotein, is expressed on neoplastic plasma cells in MM and is a recognized component of the immunosuppressive TME [42]. ISB 1342, a CD3xCD38 BiAb/BiTE, demonstrated a manageable safety profile in 24 patients with RRMM during a phase 1 dose-escalation study [43]. The development of novel targets for BiAbs in multiple myeloma is still ongoing. Of particular interest is CD138, a transmembrane proteoglycan that is highly expressed on the surface of neoplastic plasma cells [44].

These findings indicate that monotherapy with BiAbs holds tremendous therapeutic potential in MM patients (Table 1). However, the ever-present risk of antigen escape may hinder the efficacy of BiAbs. Combinatorial approaches targeting multiple antigens simultaneously have been proposed to mitigate antigen escape. The RedirecTT-1 trial enrolled RRMM patients to receive teclistamab and talquetamab, simultaneously targeting BCMA and GPRC5D, respectively [45]. A total of 63 patients received this combination therapy, achieving an ORR of 84% across all dosages [45]. Moreover, the ORR at the recommended phase 2 regimen dose was 92%. CRS and cytopenias were the most common adverse events [45]. Another potential mechanism to improve responses to BiAbs is to upregulate target antigens to MM cells. For instance, inhibiting gamma-secretase, which cleaves BCMA and releases it into the circulation, by nirogacestat has been shown to increase the expression of BCMA on multiple myeloma cells [46]. Two ongoing phase 1 studies are investigating the safety of combining nirogacestat and anti-BCMA BiAbs (NCT04722146 and NCT05090566).

Besides antigen escape, the immunosuppressive TME in MM poses challenges to the efficacy of BiAb therapy. The immune microenvironment in MM is characterized by the infiltration of T-regs and the upregulation of programmed death ligand 1 (PD-L1) on MM cells [47,48,49]. The MajesTEC-1 trial demonstrated that exhausted CD8+ T cells coupled with greater levels of T-regs resulted in lower response rates and inferior outcomes to teclistamab [22]. Moreover, the immunosuppressive TME in MM progresses in correlation with the length of disease and exposure to multiple lines of therapy [50]. Mechanistically, patients at earlier stages of their disease have more functional CD8+ cytotoxic T cells along with reduced levels of immunosuppressive T-regs [50]. Hence, the earlier utilization of BiAbs/BiTEs may improve the tumor therapy response. Several clinical trials are currently underway to explore the role of BiAbs in earlier disease stages, particularly as an adjunct to control disease activity post-autologous stem cell transplantation (NCT05623020, NCT05552222, NCT05243797, NCT05317416).

Reprogramming the TME to augment anti-tumor T-cell immunity may also improve MM responses to BiAb. In this context, IMiDs, such as lenalidomide and thalidomide, have also demonstrated the ability to enhance T-cell-directed responses against MM cells in vitro and in vivo [51]. Daratumumab, an anti-CD38 monoclonal antibody, can induce T cell expansion whilst skewing the repertoire of the TME T cells towards effector cytotoxic CD8+ T cells [52]. Clinical studies combining IMiDs and daratumumab with BiAbs are currently in their infancy, but preliminary results have shown promising outcomes. Combining teclistamab with daratumumab and lenalidomide achieved an ORR of 90% with tolerable safety profiles in a phase 1 trial [53]. The phase 1b TRIMM-2 trial combined teclistamab and daratumumab in RRMM patients and reported an ORR of 78% with manageable safety profiles [54]. Lastly, the use of immune checkpoint inhibitors, particularly agents targeting PD-L1/PD-1, can improve cytotoxic CD8+ T cell function; hence, combining ICIs with BiAbs may constitute another approach to improve responsiveness by modulating the TME of MM [55]. Figure 1 provides an overview of the BiAbs in MM and highlights the mechanisms of resistance towards them.

In summary, BiAbs represent an effective therapeutic approach for RRMM in terms of response rates and safety profiles. Overcoming antigen escape, either by combinatorial approaches employing mechanistically diverse BiAbs or pharmacologically upregulating the expression of target antigens, is an emerging area of investigation. Overcoming the immunosuppressive TME, either by earlier intervention or reprogramming it through IMiDs or ICIs, holds promise in terms of improving outcomes in RRMM patients.

## 4. Bispecific T-Cell Engagers and Antibodies in the Treatment of Acute Lymphoblastic Leukemia

Precursor B-cell acute lymphoblastic leukemia (B-ALL) is characterized by the malignant proliferation of B-lineage precursor cells in the bone marrow and peripheral blood [56]. While survival rates for adult and pediatric B-ALL patients have improved with the development of effective chemotherapeutic protocols and salvage therapies, approximately 10% of patients develop refractory disease, and there is a significant risk of relapse even after achieving initial remission—hence the need for novel therapeutic options that improve survival outcomes and facilitate minimal residual disease (MRD) clearance [57,58]. BiTEs constitute a promising strategy by reprogramming the immune system and directing T cells toward neoplastic progenitor B cells.

CD19, a critical mediator of B-cell signaling, is expressed on most cells of the B-lymphocyte lineage and is maintained during the neoplastic transition of precursor B cells in B-ALL [59]. This makes CD19 an attractive target antigen for bispecific antibody therapy. Blinatumomab is a BiTE that targets CD19 on neoplastic precursor B cells and CD3 expressed on T cells and pioneered the early development of BiAbs. The efficacy of blinatumomab was established through the phase 3 clinical trial TOWER, which compared blinatumomab to standard-of-care chemotherapy in the treatment of relapsed/refractory B-ALL [60]. The trial enrolled 405 patients, of which 271 patients received blinatumomab and 134 patients were given standard-of-care chemotherapy [60]. Blinatumomab demonstrated a significant improvement in overall survival (7.7 months vs. 4 months) along with an increase in the rates of complete remission (34% vs. 16%) [60]. However, despite achieving complete hematological remission in approximately 90% of patients, around 50% of adult patients with B-ALL still showed evidence of minimal residual disease positivity [61,62,63,64], which was strongly associated with a higher risk of relapse [65,66]. Consequently, blinatumomab was evaluated in a phase 2 clinical trial for the treatment of MRD-positive B-ALL in patients with complete hematological remission, achieving a complete MRD response in 78% of patients, leading to improved overall survival and relapse-free survival [67]. In pediatric B-ALL, phase 3 clinical trials in the pediatric population with relapsed/refractory B-ALL have demonstrated that blinatumomab is associated with improved MRD clearance and an improved likelihood of transition towards allogeneic stem cell transplantation [68,69]. The safety profile of blinatumomab is generally favorable, with primary toxicities including infection, hematologic toxicity, and neurotoxicity, and a lower incidence of CRS than other BiABs/BiTEs [60,70]. Moreover, CRS frequency can be reduced further with blinatumomab following premedication with dexamethasone and the implementation of step-up dosing. Although neurotoxicity with blinatumomab is more frequent when compared to other BiAbs/BiTEs, the clinical manifestations are transient in the majority of cases, and an improvement is noted swiftly following appropriate treatment and the interruption of blinatumomab treatment [71]. Based on these results, blinatumomab has received approval from the FDA/EMA for the treatment of relapsed/refractory B-ALL and B-ALL with MRD positivity despite complete hematological remission [72].

The success of blinatumomab has inspired investigations into its potential utility across different clinical presentations in B-ALL. Blinatumomab has been studied as an adjunct to consolidation chemotherapy in patients with MRD-negative B-ALL, demonstrating significant improvements in overall survival compared to standard consolidation alone [73]. It has also been recently evaluated in a phase 2 clinical trial in combination with induction chemotherapy in adults with Philadelphia chromosome-negative ALL, achieving MRD negativity in 92% of patients [74]. These findings suggest that blinatumomab may reduce the need for allogeneic stem cell transplantation in certain patient populations; however, more data are required from clinical trials along with greater follow-up times. Blinatumomab is currently being evaluated as a potential maintenance therapy post-allogeneic stem cell transplantation in B-ALL patients [74]. In Philadelphia chromosome t(9:22)-positive ALL, which has a poor prognosis with inferior treatment responsiveness to conventional chemotherapy, chemotherapy-free induction and consolidation regimens with blinatumomab and tyrosine kinase inhibitors (TKIs) have shown promising outcomes in two phase 2 clinical trials, indicating that this combination may potentially be superior to intensive chemotherapy, particularly in unfit patients [15,75,76,77].

In summary, revolutionary advances in B-ALL treatment have been made, spearheaded by the development of blinatumomab. Table 2 provides an overview of the main studies that outline the utility of blinatumomab in different patient populations. Nonetheless, resistance to blinatumomab poses a significant challenge to its efficacy (Figure 2). Loss of CD19 surface expression and subsequent antigen escape occurs in approximately 10–15% of patients who have relapsed following blinatumomab therapy [78,79,80]. Possible mechanisms underpinning the loss of CD19 expression include its alternative mRNA splicing, disrupted CD19 membrane trafficking, and the clonal expansion of leukemic cells that contain CD19 deletions [81,82]. Overcoming CD19 antigen loss mainly relies on identifying novel target antigens. CD22 is a novel target antigen that is expressed in the majority of leukemic blasts in B-ALL [83]. Combination therapies involving CD19 and CD22 CAR-T therapy have demonstrated promising preclinical potential and are currently being investigated in clinical trials [84,85,86]. The anti-CD22 antibody–drug conjugate inotuzumab ozogamicin (INO) has shown promising outcomes in the treatment of B-ALL, and combining INO with blinatumomab may represent a novel approach to combat antigen escape [87]. A recent phase 2 trial assessed the effect of INO with or without blinatumomab in combination with low-intensity chemotherapy amongst older adults with relapsed/refractory B-ALL, demonstrating promising outcomes regarding survival and disease clearance [88]. Another potential mechanism of resistance is the possibility of a myeloid lineage switch following blinatumomab therapy, particularly in KMT2A(MLL)-rearranged ALL, which may lead to the development of AML [89,90,91,92]. A recent preclinical study evaluated the possibility of combining anti-CD19 and anti-CD33 BiAbs to target tumor heterogeneity and prevent clonal escape [93].

Lastly, the immunosuppressive TME in ALL may promote resistance to blinatumomab [94]. A higher burden of T-regs has been associated with resistance to blinatumomab, whereas a greater presence of CD8+ effector and memory T cells and CD3+ T cells is associated with a better response to treatment [21,95]. B-ALL patients who do not respond to blinatumomab exhibit T-cell deficiency in the TME and higher levels of immune checkpoint molecules such as PD-1, TIM-3, and TIGIT compared to responders [95,96]. In agreement with these findings, a recent phase 2 clinical trial on patients with chronic lymphocytic leukemia (CLL) and Richter’s transformation to diffuse large B-cell lymphoma (DLBCL) showed that complete responders to blinatumomab expressed the lowest levels of PD-1, TIM-3, and TIGIT [97]. T-cell exhaustion may be related to exposure to multiple lines of cancer therapy before blinatumomab, as these agents are not typically used as first-line treatments, or from continuous exposure to blinatumomab, with the persistent T-cell stimulation causing subsequent exhaustion [24,25]. Accordingly, strategies to reprogram the immunosuppressive TME include treatment-free intervals, which can reduce T-cell exhaustion, and the use of ICIs such as nivolumab and pembrolizumab [24]. Results from early-stage clinical trials demonstrate that combining ICIs with blinatumomab is safe; however, efficacy results are still awaited [98,99].

## 5. Bispecific T-Cell Engagers and Antibodies in the Treatment of Non-Hodgkin’s Lymphoma

Non-Hodgkin’s lymphoma (NHL) encompasses a diverse group of lymphoproliferative neoplasms with varying grades of progression and severity [100]. Among the numerous NHL subtypes, indolent follicular lymphoma (FL) and diffuse large B-cell lymphoma (DLBCL) are the most common [100]. The introduction of the anti-CD20 monoclonal antibody rituximab has significantly improved the prognosis of B-cell NHL. However, a significant number of patients develop relapsed and/or refractory disease that does not respond to conventional chemotherapy [101]; hence, the need arises for novel treatment strategies, such as those that harness T-cell-mediated anti-neoplastic activity. CAR-T therapy has demonstrated remarkable efficacy in the treatment of relapsed/refractory NHL, but urgent intervention is required for patients with rapidly progressive disease [102].

BiAbs targeting multiple effector cell surface markers (CD3, CD16a, 4-1BBL, CD28, CD47) and B-cell antigens (CD19, CD20, CD22, CD37, CD79b) have been developed for NHL treatment [103]. CD20 is a critical B-cell surface antigen that is expressed on approximately 90% of malignant B cells but not on hematopoietic stem cells, minimizing the risk of myelosuppression [104,105]. These characteristics render CD20 an attractive target antigen for BiAbs in NHL [106]. Currently, several CD3xCD20 BiAbs, including glofitamab, mosunetuzumab, epcoritamab, odronextamab, and Igm-2323, have shown significant activity in the treatment of both indolent and aggressive NHL subtypes, including FL, DLBCL, transformed follicular lymphoma (tfFL), primary mediastinal large B-cell lymphoma (PMBCL), mantle cell lymphoma (MCL), and Richter’s transformation, in phase 1 and 2 clinical trials (Table 3). In patients with relapsed/refractory FL (RRFL), mosunetuzumab and odronextamab have demonstrated compelling efficacy, achieving complete response rates of 60% and 75%, respectively [107,108]. In DLBCL patients, glofitamab, odronextamab, and epcoritamab displayed similar complete response rates of 37–39% [109,110,111,112]. However, when considering the durability of responses, glofitamab showed better results, with 70% of patients still in complete remission after 18 months, compared to 48% with odronextamab [110,112].

Regarding adverse events, low-grade CRS is the most common side effect associated with CD20xCD3 antibodies, with grade ≥3 occurring rarely (Table 3). CRS events usually occur during the first cycle of treatment, and their severity can be attenuated by step-up dosing, premedication with steroids, and the administration of a B-cell-depleting agent [107,108,110,111,113,114]. Other adverse events, such as pyrexia, neutropenia, anemia, and electrolyte changes, are transient and clinically insignificant. Importantly, CD20xCD3 BiAbs are associated with a lower incidence and severity of immune effector cell-associated neurotoxicity syndrome (ICANS) than CAR-T therapy [110,115]. Furthermore, infections were common but varied between BiAbs, which could be attributed to different rates of neutropenia among BiAbs (38% with glofitamab vs. 21.7% with epcoritamab) [110,111]. Similarly, hypogammaglobulinemia associated with CD20xCD3 BiAbs may predispose patients to infection.

**Table 3 cancers-15-04550-t003:** Phase 1 and 2 studies evaluating the safety and efficacy of CD20xCD3 bispecific antibodies in different subtypes of non-Hodgkin’s lymphoma.

BiAB, Trial	BiAB Structure	N	Design	ORR, CR (%)	CRS (All Grade, ≥Grade 3) %	ICANS %	Infections %
Mosunetuzumab (Ph2, NCT02500407) [107]	IgG1, humanized	90	IV, 21-day cycles, step-up dosing (1/2/60/60 mg) then 30 mg onwards. Pts achieving a CR by cycle 8 completed treatment; those with a partial response or stable disease received 17 cycles total	RRFL 77.8, 60.0	44.0, 2.0	NR	NR
Odronextamab (Ph2, NCT03888105) [108]	Fully human IgG4-based	96	IV, 21-day cycles, step-up dosing in two regimens (1/20 mg or 0.7/4/20 mg) then 80 mg till cycle 4. Followed by 160 mg maintenance every 2 weeks till disease progression or unacceptable toxicity	RRFL 81.0, 75.0	51.0, 0.0	0.0 in 0.7/4/20 regimen 3.0 in 1/20	NR
Odronextamab (Ph2, NCT03888105) [112]	Fully human IgG4-based	121	IV, 21-day cycles, step-up dosing in two regimens (1/20 mg or 0.7/4/20 mg) then 160 mg till cycle 4. Followed by 320 mg maintenance every 2 weeks till disease progression or unacceptable toxicity	RR DLBCL 53.0, 37.0	53.0, 0.0	4.0 in 0.7/4/20 regimen 6.0 in 1/20	NR
Epcoritamab (Ph2, NCT03625037) [111]	IgG1, humanized	157	SQ, 28-day cycles, once weekly step-up doses in weeks 1–3 of cycle 1, then full doses once weekly through cycle 3, once every 2 weeks in cycles 4–9, and once every 4 weeks in cycle 10 and thereafter, until disease progression or unacceptable toxicity	RR DLBCL 63.0, 39.0	49.7, 2.5	6.4 (one death)	45.2
Glofitamab (Ph2, NCT03075696) [109]	2:1 configuration with bivalency to CD20	154	Pre-treatment with 1000 mg obinutuzumab, followed by IV glofitamab 7 days later, 21-day cycles, two step-up doses (2.5/10 mg) then 30 mg for 12 cycles.	RR DLBCL 58.0, 38.0	64.0, 4.0	8.0	59.0
Mosunetuzumab (Ph1/2, NCT02500407) [116]	IgG1, humanized	89	SQ, 21-day cycles, step-up dosing, 3 groups (5/15/45 mg, 5/45/45 mg, 5/45/90/90/45 mg) then 45 mg onwards. Pts achieving a CR by cycle 8 completed treatment; those with a partial response or stable disease received 17 cycles total	iNHL 82.0, 64.0 aNHL 36.0, 20.0	27.0, 0.0	3.0	14.0 grade 3/4
Igm-2323 (Ph1, NCT04082936) [117]	Ten binding domains for CD20; one binding domain for CD3	29	IV on days 1, 8, and 15 of 21-day cycles until disease progression	(FL *n* = 11) (DLBCL *n* = 13) (MCL *n* = 3) (MZL *n* = 2) 34.8, 21.7	20.7, NR	0.0	NR

BiAb = bispecific antibody. CRS = cytokine release syndrome. ICANS = immune effector cell-associated neurotoxicity syndrome. ORR = overall response rate. CR = complete response. NR = not reported. DLBCL = diffuse large B-cell lymphoma. iNHL = indolent NHL. aNHL = aggressive NHL. MCL = mantle cell lymphoma. MZL = marginal zone lymphoma. SQ = subcutaneous.

CD19 is another potential target antigen in NHL due to its ubiquitous expression on B cells, including neoplastic B cells [118]. CD19-targeting BiAbs, such as blinatumomab, AFM11, duvortuxizumab, and Tnb-486, have been evaluated in NHL treatment [119,120,121,122]. Blinatumomab’s success in the treatment of ALL paved the way for efforts to explore its efficacy in NHL. Blinatumomab has shown substantial efficacy for the treatment of NHL in phase I and II clinical trials but is associated with a high rate of potentially severe neurological events [119,123]. The high frequency of neurological events, coupled with blinatumomab’s narrow half-life, necessitating continuous infusions, has halted any further development in its use for NHL [119,123]. Similarly, phase I studies assessing AFM11 and duvortuxizumab were discontinued due to neurotoxicity concerns [120,122]. Tnb-486, a novel CD3xCD19 BiAb, demonstrated a complete response in 91% of RRFL patients in a phase 1 trial and had a lower incidence of ICANS and CRS compared to blinatumomab [119,121,123]. Mechanistically, the lower incidence of ICANS and CRS associated with Tnb-486 is likely due to its unique anti-CD3 moiety, designed to bind CD3 on T cells with low affinity, thereby attenuating the release of pro-inflammatory cytokines [124]. The higher incidence of ICANS observed with CD19xCD3 BiAbs when compared to CD3xCD20 BiAbs may be due to potential on-target off-tumor toxicity associated with targeting CD19, which is expressed on the pericytes and vascular smooth muscles that line the blood–brain barrier (BBB); therefore, the use of anti-CD19 BiAbs such as blinatumomab may impair the integrity of the BBB [125].

The CD20xCD3 BiAbs mosunetuzumab, glofitamab, and epcoritamab have received accelerated FDA approval for specific NHL subtypes due to their substantial efficacy. However, factors impairing the efficacy of CD20xCD3 BiAbs require further exploration. For example, antigen escape resulting from reduced CD20 expression has been observed in a significant number of NHL patients treated with rituximab and was associated with an inferior prognosis [126]. Loss of CD20 expression has also been associated, deemed to be a potential contributor towards resistance to the CD20xCD3 BiAb mosunetuzumab [121]. Upregulating CD20 expression on NHL cells may, therefore, constitute a potential mechanism to enhance the efficacy of CD20xCD3 BiAbs. In this regard, gemcitabine can upregulate CD20 on DLBCL cells in vitro, which enhances the antitumor activity of rituximab [127]. A recent phase Ib/II trial demonstrated that epcoritamab + GemOx (gemcitabine, oxaliplatin) in RR DLBCL displayed a higher ORR than epcoritamab monotherapy (92% vs. 63%) [111,124]. Moreover, ameliorating antigen escape through targeting multiple antigens simultaneously in NHL may provide a novel approach to enhance their efficacy [113,128]. The antibody–drug conjugate polatuzumab vedotin (PV) targets CD79b, an antigen that is expressed on the majority of malignant B cells in NHL [129]. Two phase Ib/II clinical trials assessing the efficacy and safety of glofitamab and mosunetuzumab in combination with PV have demonstrated promising results, with ORRs of 80% with glofitamab and 72% with mosunetuzumab in RR DLBCL [113,128]. The ORRs observed with this combination appear to be superior to those reported with both glofitamab monotherapy (ORR 58.0%) and mosunetuzumab monotherapy (ORR 42.0%) in RR DLBCL [109,130]. Additionally, results from a phase Ib study demonstrated that glofitamab + Pola-R-CHP (PV, rituximab, cyclophosphamide, doxorubicin, prednisone) in patients with treatment-naive DLBCL demonstrated an ORR of 100% in patients who completed the treatment cycle [121]. These combination regimens have also demonstrated remarkable efficacy in treatment-naive patients in numerous ongoing phase 3 trials comparing their efficacy to standard-of-care treatment [131,132].

Chronic exposure to BiAbs results in continuous T-cell stimulation and subsequent exhaustion, impairing the efficacy of BiAbs/BiTEs [133]. In vitro studies have shown that continuous exposure to BiAbs impairs the T-cell-mediated lysis of neoplastic cells [23]. Exhausted T cells, characterized by the increased expression of inhibitory checkpoint molecules such as PD-1 [134], correlated with disease progression in DCLBL patients receiving glofitamab [135]. Combining BiABs with IMiDs, such as lenalidomide, can enhance T-cell activation via B7-CD28-mediated signaling and reduce T-cell exhaustion and surface PD-1 expression [136]. Phase 1–2 clinical trials combining epcoritamab with lenalidomide + rituximab (R2) in patients with relapsed/refractory FL have demonstrated improved efficacy for the combination regimen than epcoritamab monotherapy (complete response rates of 86% vs. 50%, respectively) [114,137]. Multiple clinical trials evaluating the efficacy of a BiAb and lenalidomide combination for RRFL are underway, with promising phase 1 safety results, but are yet to be conducted in NHL [138,139,140,141]. Another strategy is to activate co-stimulatory receptors, such as CD28 and 4-1BB on T cells, which improve T-cell activation, expansion, and survival [142,143]. Preclinical evidence demonstrated that the antineoplastic capacity of odronextamab was enhanced through its combination with REGN5837, a BiAb that cross-links CD28 on T cells with CD22 on tumor cells. REGN5837 was capable of reactivating exhausted T cells, expanding the intra-tumoral population of T cells, and promoting T-cell persistence, resulting in increased tumor lysis [144]. Additionally, RO7227166, a novel CD19 × 4-1BBL costimulatory BiAb, enhanced the anti-tumor efficacy of glofitamab [145]. Thus, future efforts should be directed towards exploring novel strategies to enhance the T-cell-engaging capacity of BiAbs to dampen the risk of therapeutic resistance facilitated by the immunosuppressive microenvironment of NHL.

## 6. Bispecific T-Cell Engagers and Antibodies in the Treatment of Acute Myelogenous Leukemia

Acute myelogenous leukemia (AML) is characterized by the infiltration of immature leukemic cells in the bone marrow and their accumulation in peripheral blood [146]. Survival rates and outcomes of AML have not improved substantially over the past few decades [146]. Chemotherapy followed by allogeneic stem cell transplantation is the standard of care for AML, but many patients develop relapse or treatment-refractory disease [147]. T-cell-engaging immunotherapies, such as BiAbs, offer a novel approach to target chemotherapy-resistant AML tumor cells. However, the application of BiAbs in AML faces challenges due to the limited target antigens that are ubiquitously expressed on malignant AML cells [148]. Additionally, the overlapping expression of target antigens between malignant AML cells and hematopoietic stem cells raises concerns about on-target off-tumor toxicities, particularly hematologic toxicity and cytopenias.

Current bispecific T-cell engagers in AML target many antigens, including CD33, CD123, CLL-1, and FLT-3 (Figure 3) [149]. CD33, a glycoprotein expressed on immature myeloid blasts and leukemic stem cells, has garnered significant interest. CD33 is of particular interest due to its expression on the majority of immature myeloid blasts and leukemic stem cells [149,150]. High CD33 expression correlates with adverse cytogenetic profiles and poor outcomes [151]. However, CD33 is also expressed on hematopoietic stem cells, which increases the risk of myelosuppression [152]. Gemtuzumab ozogamicin, an anti-CD33 antibody–drug conjugate, is approved for relapsed/refractory AML in CD33+ adults [153,154]. Clinical trials evaluating the efficacy of anti-CD33 bispecific T-cell engagers such as AMG 673, AMG-330, and GEM333 have been terminated despite promising preliminary results; however, there are two CD33xCD3 BiAbs in clinical development after the completion of initial phase 1 studies (JNJ-67561244 and AMV564).

CD123, the low-affinity binding subunit of the IL-3 receptor, has emerged as another target antigen for BiTE therapy in AML. CD123 is widely expressed on leukemic stem cells and myeloid blasts and correlates with disease severity and therapy resistance [155,156,157]. Flotetuzumab, a CD123xCD3 dual-affinity retargeting protein (DART), demonstrated anti-leukemic activity with a manageable toxicity profile in relapsed/refractory AML patients [17]. XmAb14045, another CD123xCD3 targeting BiAb, also showed anti-leukemic activity in relapsed/refractory AML patients and is currently being assessed in a phase II clinical trial (NCT05285813) [158]. Other CD123xCD3 BiAbs, such as APVO436 and MGD024, are being evaluated in ongoing phase 1 clinical trials after demonstrating anti-leukemic activity in preclinical studies [159,160].

CLL-1 (CLEC12A) is another potential target antigen for BiTE/BiAb therapy in AML due to its expression on leukemic stem cells and myeloid blasts but not hematopoietic stem cells [161,162,163]. However, CLL-1 has low expression levels on cell surfaces, potentially impairing the antibody activity [162]. MCL-117 is a CLL-1xCD3 bispecific T-cell-engaging antibody that has shown promise in preclinical studies but did not yield optimal clinical responses (NCT03038230) [164,165]. FLT-3, a receptor-type tyrosine kinase, is expressed on leukemic stem cells and myeloid blasts and represents a promising target antigen for BiAbs/BiTE therapy in AML [166,167,168]. FLT-3 inhibitors such as midostaurin and gilteritinib are currently approved for use in FLT-3-mutated AML patients [169,170]. However, FLT-3 expression on hematopoietic stem cells and its limited presence across different AML subtypes pose challenges [171]. CLN-049, a CD3xFLT3 BiTE, demonstrated anti-leukemic activity in preclinical studies and is currently being evaluated in a phase 1 trial (NCT05143996) [172].

BiTEs hold promise in AML treatment, but clinical trials are currently in their infancy (Table 4). There are also several limitations, such as the toxicity of BiAbs/BiTEs, which may hinder the utilization of BiAb/BiTE therapy in AML. Adverse events, such as CRS and cytopenias, are associated with BiTE therapy, but step-up dosing and premedication with steroids have shown to be effective in reducing their severity [17,158]. Strategies to reduce CRS, such as premedication or design modification, should be explored further [173]. Another limitation is potential antigen loss or insufficient expression of the targeted antigen on cell surfaces. For example, single nucleotide polymorphisms (SNP) in the CD33 splicer enhancer region can alter the antibody-binding domain of CD33, resulting in resistance to gemtuzumab ozogamicin [174]. Combining BiAbs targeting multiple antigens on AML cells may overcome therapy resistance due to target antigen loss or structural alterations. The immunosuppressive TME in AML contributes to therapy resistance [175,176,177]. For instance, myeloid-derived suppressor cells (MDSCs) expressing CD14 and CD33 may curb effective anti-tumor responses [178,179]. Additionally, the upregulation of immune checkpoints such as PD-L1 on T cells has been shown to correlate with an impaired anti-tumor T-cell response in AML [180,181,182]. The CD33xCD3 BiAbs AMG330 and AMV564 have demonstrated anti-leukemic activity by activating T cells and modulating MDSCs [183,184,185]. Notably, AMG330 induces potent inflammatory cytokine responses, resulting in the upregulation of PD-L1 on AML cells and subsequent immune evasion [150]. These results indicate that combining BiAbs/BiTEs with immune checkpoint inhibitors such as anti-PD-1 may provide a novel approach to augment bispecific antibody therapy in AML and attenuate treatment resistance.

## 7. Overview of the Toxicities Associated with the Use of Bispecific Antibodies in the Treatment of Hematologic Malignancies

The previous sections have expounded on the therapeutic potential of BiAbs/BiTEs in hematological malignancies. However, ensuring the safety and tolerability of these therapeutic modalities is of paramount importance to incorporate them into treatment protocols for hematological malignancies. The primary adverse events associated with BiAb and BiTE treatment include CRS, infections, hematological toxicity, and neurotoxicity (Figure 4). This section provides an overview of these toxicities associated with BiAbs and BiTE therapy and explores strategies to alleviate these toxicities and minimize their impact.

### 7.1. Cytokine Release Syndrome (CRS)

CRS is characterized by an exaggerated inflammatory response with elevated levels of cytokines interleukin-2 (IL-2), interleukin-6 (IL-6), interferon-gamma (IFN-γ), and tumor necrosis factor-alpha (TNFα) [186,187]. The clinical manifestations of CRS are variable, ranging from mild fever and malaise to severe hypotension and hypoxia [188]. The severity of CRS is graded according to guidelines from the American Society for Transplantation and Cellular Therapy (ASTCT) [189]. Grades 1 and 2 are more common and characterized by non-life-threatening symptoms, whereas grades 3 and 4 require urgent intervention due to the life-threatening nature of symptoms. For instance, a recent meta-analysis of 53 studies found that the rate of CRS in patients treated with BiAbs was 67%, but the rate of severe (grade 3 or 4) CRS was 0.2% [190]. In the context of BiAb and BiTE therapy, CRS occurs due to T-cell activation via the CD3 component of BiTEs/BiAbs [191]. CRS tends to occur primarily during the first cycle of treatment. Additionally, comparing intravenous and subcutaneous administrations of BiAbs/BiTEs, CRS tends to manifest on the first day of intravenous administration compared to the second day of subcutaneous administration [23].

Therefore, intensive monitoring is essential for patients receiving BiAb/BiTE therapy, particularly during the initial 48 hours of dose administration. Step-up dosing has been shown to mitigate the release of inflammatory cytokines and reduce the duration and intensity of CRS [192]. Additionally, the use of BiAbs/BiTEs with a lower affinity for CD3 may ameliorate CRS [192]. Pretreatment with immunosuppressive/immunomodulatory drugs can also attenuate CRS both in vitro and in vivo [193].

When patients develop CRS, immediate supportive care is needed, followed by admission to intensive care units. Supportive care includes the maintenance of normoxia and administration of fluids/antipyretics. Furthermore, the administration of steroids and/or IL-6-blocking mAb tocilizumab can significantly alleviate both the duration and severity of CRS [188].

### 7.2. Infections

It is vital to closely monitor patients receiving BiAb/BiTE therapy for signs of infection. Patients should be screened for opportunistic or reactivation infections such as cytomegalovirus and Epstein–Barr virus [194]. A pooled analysis of 1185 patients receiving BiAb therapy identified a 50% rate of infections, of which 24.5% were graded as severe [195]. The prevalence of hypogammaglobulinemia was reported to be 75.3%. Furthermore, 25.5% of the total deaths were attributed to infections [195]. The incidence of infections in the context of BiAb/BiTE therapy is often multifactorial. Patients with hematological malignancies who receive BiAb/BiTE therapy often have refractory disease with prior exposure to multiple lines of therapy, increasing their risk of infection. Moreover, patients with active hematological malignancies are often neutropenic due to impaired hematopoiesis. Anti-BCMA BiAb/BiTEs in multiple myeloma and anti-CD20 BiAb/BiTEs in non-Hodgkin’s lymphoma impair B-cell function, resulting in profound hypogammaglobulinemia [196,197]. Lastly, continuous T-cell stimulation by BiAbs/BiTEs may promote T-cell exhaustion, increasing the susceptibility to infections [24]. Preventative strategies to minimize the infection risk in BiAb/BiTE therapy include prophylactic IVIG and antimicrobials (antibiotics, antivirals, and antifungals) [194,195,196].

### 7.3. Hematologic Toxicity

Patients treated with BiAbs/BiTEs are predisposed to the development of hematological toxicities, including anemia, thrombocytopenia, and neutropenia. The exact mechanism behind the development of these toxicities remains unknown but may be linked to therapy-induced pro-inflammatory cytokine release and/or the impairment of hematopoiesis [198,199]. Supportive measures, such as the transfusion of blood products and granulocyte colony-stimulating factors, should be considered to improve hematologic parameters and reduce the infection risk.

### 7.4. Neurotoxicity

Neurotoxicity in the context of BiAb/BiTE therapy can arise either from CRS or as a consequence of immune effector cell-associated neurotoxicity syndrome (ICANS) [200]. The symptoms of neurotoxicity due to CRS or ICANS are variable and may include seizures, confusion, tremors, dysphasia/aphasia, and ataxia [200]. The severity of ICANS is graded based on the ASTCT guidelines [189]. Strategies to alleviate the risk of ICANS are similar to those employed in CRS, including steroids and tocilizumab. Notably, blinatumomab carriers the highest risk of neurotoxicity, likely due to CD19 co-expression in neural tissue, necessitating a high index of suspicion for ICANS for patients receiving blinatumomab [71,125].

## 8. Conclusions and Future Directions

BiAbs/BiTEs have transformed the treatment landscape for advanced hematological malignancies, with several approved BiAbs/BiTEs showing promising efficacy and favorable safety profiles. Additionally, early-stage clinical trials of numerous other BiTEs have demonstrated encouraging anti-neoplastic activity, raising optimism for their approval in the coming years.

However, the efficacy and tolerability of certain BiAbs/BiTEs warrant further exploration. Antigen escape is a major resistance mechanism to BiAb/BiTE therapy. Elucidating additional tumor-associated target antigens and exploring combinatorial, multi-antigenic BiAb/BiTE may counteract antigen escape. Additionally, the immunosuppressive TME in hematological malignancies is a significant contributor to BiAb/BiTE resistance. Investigating strategies to normalize the tumor microenvironment, such as immune checkpoint inhibitors and immunomodulatory agents, may enhance the efficacy of BiAbs/BiTEs and mitigate therapeutic resistance.

Presently, much of the BiAb/BiTE research in hematological malignancies focuses on patients with relapsed/refractory disease, but it is plausible that patients in the early stages of their disease may respond to BiAbs/BiTEs more favorably due to their lower tumor burden and a TME more conducive to anti-tumor immune responses. We have already discussed how T-cell dysfunction consequent to exposure to multiple lines of cancer therapy has been associated with resistance to BiABs, early-stage disease being associated with a better therapeutic response to BiTEs, and clinical trials being underway to assess the efficacy of BiAbs/BiTEs in early-stage hematologic malignancies. Exploring novel BiAb engineering strategies such as trispecific antibodies—which target more tumor antigens and minimize the risk of antigen escape—is also important. Incorporating natural killer cell engagers may also synergize with bispecific T-cell engagers and potentially enhance the anti-tumor immune response.

In conclusion, the field of bispecific T-cell engagers holds tremendous therapeutic potential, and we eagerly anticipate further progress from future preclinical studies and clinical trials. The continued advancement of these therapies is expected to have a significant impact on the treatment of hematological malignancies, bringing hope to patients and healthcare professionals alike.

## Figures and Tables

**Figure 1 cancers-15-04550-f001:**
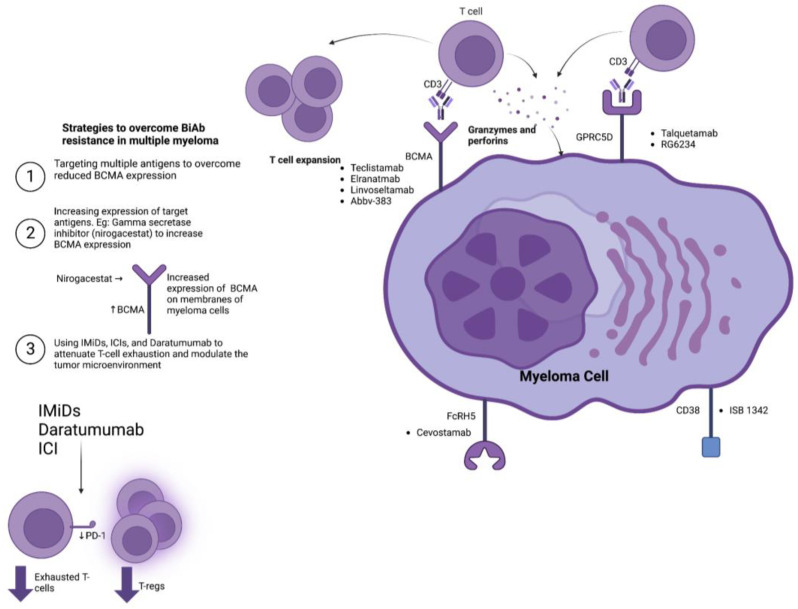
This figure depicts the main bispecific antibodies in ongoing clinical trials for the treatment of multiple myeloma. The bispecific antibodies are outlined according to their target myeloma-associated antigen, including BCMA, GPRC5D, FcRH5, and CD38. The figure also outlines the key strategies to overcome bispecific antibody resistance. One of the key strategies includes the synergistic combination of bispecific antibodies with other bispecific antibodies, monoclonal antibodies, or antibody–drug conjugates in order to target multiple antigens simultaneously. The second strategy to overcome resistance is relevant to the main class of bispecific antibodies in multiple myeloma, i.e., BCMA-targeting bispecific antibodies. This strategy involves using gamma-secretase inhibitors such as nirogacestat to prevent the cleavage of membrane-bound BCMA into soluble BCMA, thereby increasing the expression of BCMA on the surfaces of myeloma cells. Finally, the third strategy involves the use of immunomodulatory agents, immune checkpoint inhibitors, and daratumumab to modulate the tumor microenvironment and shift the repertoire of T cells by reducing the number of immunosuppressive regulatory T cells and exhausted T cells.

**Figure 2 cancers-15-04550-f002:**
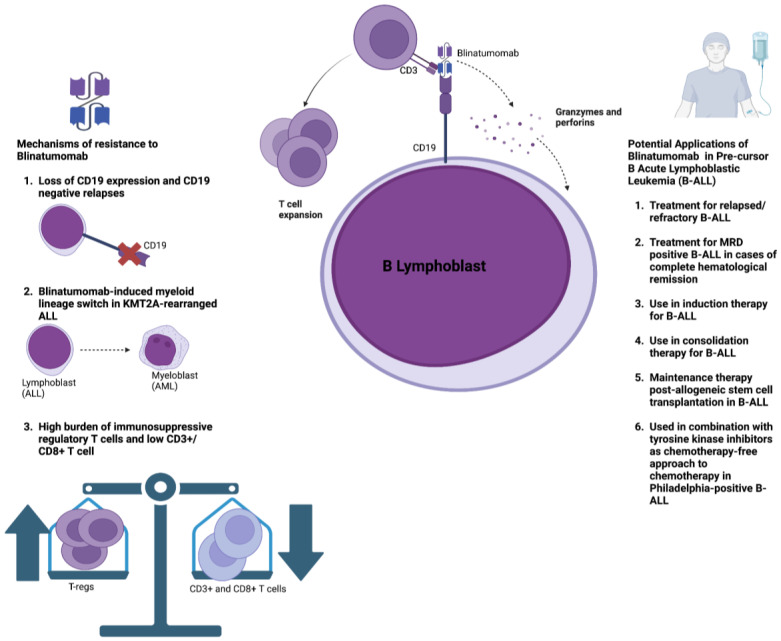
This figure outlines the mechanism of action of blinatumomab, the main bispecific T-cell engager utilized in the treatment of acute lymphoblastic leukemia. Blinatumomab binds to the CD19 antigen expressed on neoplastic B lymphoblasts along with the CD3 receptor expression on T cells, resulting in subsequent T-cell-mediated lysis of the leukemic cell. The figure also outlines some of the potential applications of blinatumomab therapy in acute lymphoblastic leukemia. This includes its use for relapsed/refractory disease, in addition to its incorporation in induction and consolidation regimens for different subtypes of acute lymphoblastic leukemia. The figure also illustrates the key mechanisms of resistance associated with blinatumomab therapy, including (1) antigen escape and loss of CD19 expression; (2) myeloid lineage switch after blinatumomab therapy, which has been reported in cases of KMT2A(MLL)-rearranged acute lymphoblastic leukemia, resulting in the development of acute myelogenous leukemia; (3) the immunosuppressive microenvironment in acute lymphoblastic leukemia is associated with an increased percentage of regulatory T cells along with a lower frequency of CD8+/CD3+ T cells, thereby facilitating resistance to blinatumomab.

**Figure 3 cancers-15-04550-f003:**
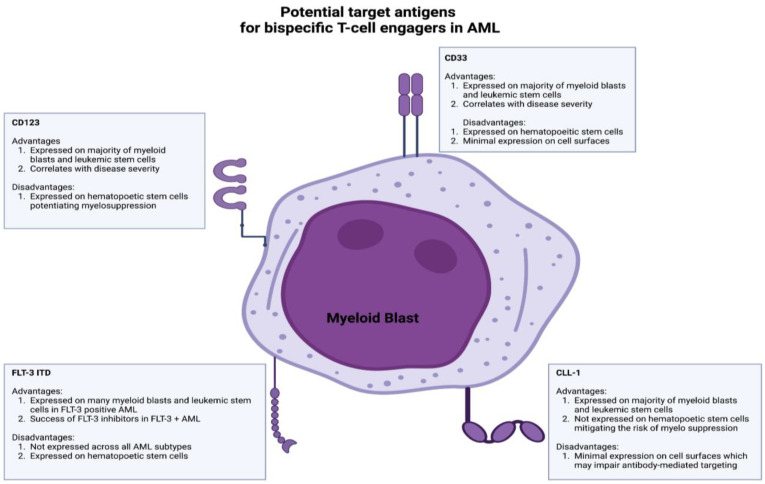
This figure outlines the advantages and disadvantages of the potential target antigens in the development of bispecific antibodies for the treatment of acute myelogenous leukemia. These target antigens, which are expressed on leukemic stem cells and myeloblasts, include CD33, CD123, CLL-1, and FLT-3.

**Figure 4 cancers-15-04550-f004:**
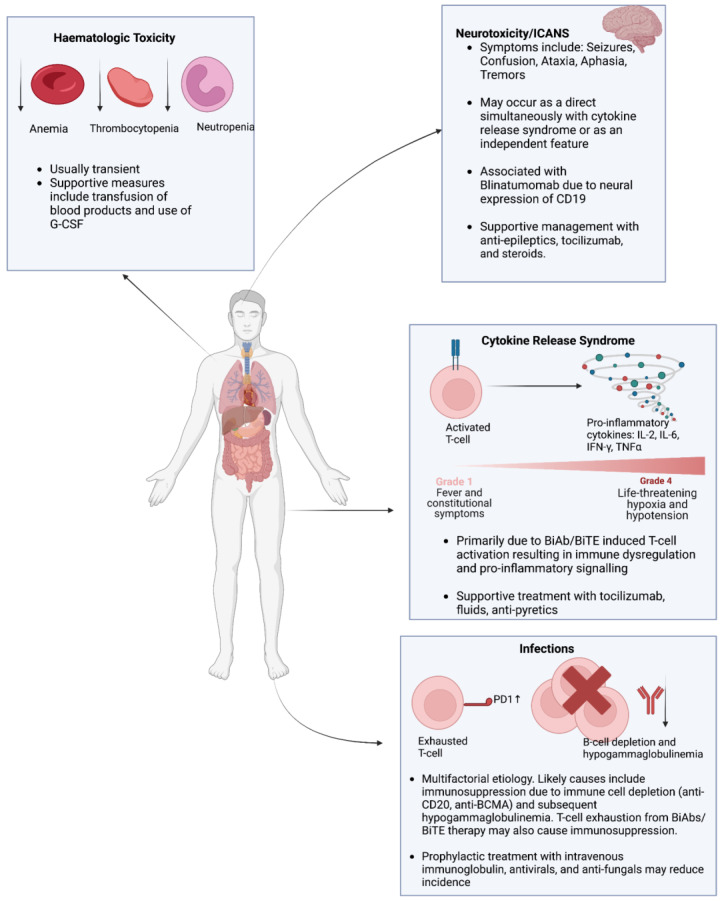
This figure outlines the features of primary toxicities associated with bispecific T-cell engaging therapy in hematological malignancies. These primary toxicities include neurotoxicity/immune effector cell-associated neurotoxicity syndrome, cytokine release syndrome, infections, and hematologic toxicity.

**Table 1 cancers-15-04550-t001:** Phase 1 and 2 studies evaluating the safety and efficacy of bispecific antibodies targeting BCMA, GPRC5D, and FcRH5 in multiple myeloma.

BiAB, Trial	Targets	BiAB Structure	N	Design	ORR, CR (%)	CRS (All Grade, ≥Grade 3) %	ICANS (%)	Infections (%)
Teclistamab (Ph1-2, NCT04557098) [32]	BCMAxCD3	Humanized IgG Fc	165	SQ, weekly injection at dose of 1.5 mg/kg. Step-up doses of 0.06 mg and 0.3 mg per kilogram.	63.0, 39.4	72.1, 0.6	3.0	76.4
Elranatamab (Ph2, NCT04649359) [33]	BCMAxCD3	Humanized IgG2a	123	SQ, weekly injection at a dose of 76 mg for a 28-day cycle. Two step-up doses at 12 mg and 32 mg.	61.0, 27.6	56.3, 0.0	3.4	61.8
Linvoseltamab (Ph2, NCT03761108) [34]	BCMAxCD3	Fc Fab arms	252	Two cohorts received doses of 50 mg and 200 mg, respectively. IV, with two step-up doses. A protocol amendment allowed pts who progressed at 50 mg to dose escalate to 200 mg.	50 mg cohort: 50.0, 20.2 200 mg cohort: 64.0, 24.1	50 mg cohort: 53.0, 1.0 200 mg cohort: 37.0, 2.0	Grade 3 or 4 50 mg cohort: 1.0 200 mg cohort: 2.0	50 mg cohort: 59.0 200 mg cohort: 43.0
Abbv-383 (Ph1, NCT03933735) [35]	BCMAxCD3	IgG4 Fc	124	IV, once every 3 weeks. Doses of 40 mg and 60 mg for escalation and expansion cohorts.	57.0, 29.0	40 mg cohort: 83.0, 0.0 60 mg cohort: 72.0, 2.0	NR	40 mg cohort: 50.0 60 mg cohort: 43.0
Talquetamab (Ph1, NCT03399799) [36]	GPRC5DxCD3	Humanized IgG4	232	102 patients IV weekly or every other week at doses from 0.5 to 180 μg per kilogram of body weight. 130 patients SQ weekly, every other week, or monthly at doses from 5 to 1600 μg per kilogram.	At SQ doses of 405 μg/kg: 70.0, 23.0 and 800 μg/kg: 64.0, 23.0	At SQ doses of 405 μg/kg: 77.0, 3.0 and 800 μg/kg: 80.0, 0.0 At IV doses: 49.0, 5.0	NR	NR
Cevostamab (Ph1, NCT03275103) [41]	FcRH5xCD3	Humanized IgG1	160	IV administration in 21-day cycles. Two step-up doses.	At 160 mg dose: 54% At 90 mg dose: 36.7	80.0, 1.3	NR	42.5, 18.8

BiAb = bispecific antibody. CRS = cytokine release syndrome. ICANS = immune effector cell-associated neurotoxicity syndrome. ORR = overall response rate. CR = complete response. NR = not reported. SQ = subcutaneous. IV = intravenous.

**Table 2 cancers-15-04550-t002:** Selected clinical trials evaluating the efficacy of blinatumomab in different patient populations.

First Author, Year	Phase	N	Study Design and Patient Population	Outcomes	Adverse Events
Kantarjian et al. 2017 [60]	3	405	Heavily pretreated relapsed/refractory Philadelphia-negative B-ALL. Randomized 2:1 comparison between blinatumomab and standard-of-care chemotherapy.	Median overall survival in blinatumomab group 7.7 months vs. 4.0 months in standard-of-care group. Complete hematologic remission in 34% in the blinatumomab group vs. 16% in the standard-of-care group.	Infection in 34.1% of the blinatumomab group vs. 52.3% in the standard-of-care group. Neurotoxicity in 9.4% in the blinatumomab group vs. 8.3% in the standard-of-care group.
Gökbuget et al. 2018 [67]	2	116	Open-label, single-arm study, adults with B-cell precursor ALL in hematologic complete remission with MRD (≥10^−3^).	MRD clearance in 78% of patients. Relapse-free survival at 18 months 54%. Median overall survival of 36.5 months.	Cytokine release syndrome in 3%. Neurotoxicity grade 3 in 10%, grade 4 in 3%.
Brown et al. 2021 [69]	3	208	Ages 1–30 years with first relapse B-ALL. Randomized between 2 cycles of blinatumomab and 2 cycles multi-agent chemotherapy.	2-year disease free survival 54.4% in the blinatumomab group vs. 39.0% in the chemotherapy group. 2-year overall survival 71.3% in the blinatumomab group vs. 58.4% in the chemotherapy group.	Infection in 15.0% of the blinatumomab group vs. 65.0% in the chemotherapy group.
Litzow et al. 2022 [73]	3	224	Patients with negative MRD (<0.01%) post-induction therapy were randomized to either receive conventional consolidation chemotherapy or blinatumomab in addition to conventional consolidation.	Upper boundary for efficacy analysis was crossed in favor of blinatumomab, with a significant improvement in overall survival in favor of blinatumomab arm. Median overall survival not reached vs. 71.4 months, hazard ratio 0.42, *p* = 0.003.	NR
Salek et al. 2022 [74]	2	29	Single cycle of blinatumomab followed by high-dose chemotherapy in induction therapy for Philadelphia-negative adult ALL.	93% of patients achieved complete hematological remission after induction, of which 52% were complete molecular remissions.	Febrile neutropenia in 15%, and hepatotoxicity in 11%. No neurotoxicity observed.
Foà et al. 2020 [15]	2	63	Philadelphia-positive ALL patients. Single-arm trial in which Dasatinib plus glucocorticoids were administered, followed by two cycles of blinatumomab.	Complete remission achieved in 98%. At median follow-up of 18 months, overall survival was 95% with disease-free survival of 88%.	Grade ≥ 3 adverse events included cytomegalovirus reactivation in 6 patients, neutropenia in 4 patients, and neurotoxicity in one patient.

**Table 4 cancers-15-04550-t004:** Past and present clinical trials evaluating the use of bispecific T-cell engagers and antibodies in acute myelogenous leukemia.

Trial ID	Antibody Name	Targets	Patient Population	Phase	Primary Outcomes	Status
NCT02520427	AMG 330	CD33xCD3	Relapsed/refractory AML, MDS	1	Safety	Terminated
NCT03224819	AMG 673	CD33xCD3	Relapsed/refractory AML	1	Safety	Terminated
NCT03516760	GEM333	CD33xCD3	Relapsed/refractory AML	1	Safety	Terminated
NCT03915379	JNJ-67571244	CD33xCD3	Relapsed/refractory AML, MDS	1	Safety and efficacy	Completed
NCT03144245	AMV564	CD33xCD3	Relapsed/refractory AML	1	Safety and efficacy	Completed
NCT04582864	Flotetuzumab	CD123xCD3	Relapsed/refractory AML	2	Efficacy	Recruiting
NCT05285813	XmAb14045	CD123xCD3	Relapsed/refractory AML, MDS	2	Efficacy	Recruiting
NCT03647800	APVO436	CD123xCD3	Relapsed/refractory AML, MDS	1	Safety	Recruiting
NCT05362773	MGD024	CD123xCD3	Relapsed/refractory AML, MDS, Hodgkin’s lymphoma, B-cell leukemia, hairy cell leukemia, CML, systemic mastocytosis	1	Safety	Recruiting
NCT02715011	JNJ-63709178	CD123xCD3	Relapsed/refractory AML	1	Safety	Completed
NCT03038230	MCLA-117	CLL-1xCD3	Relapsed/refractory AML	1	Safety	Halted
NCT05143996	CLN-049	FLT-3xCD3	Relapsed/refractory AML, MDS	1	Safety	Recruiting

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
