# Peer review of "Bispecific Antibodies in Hematological Malignancies: A Scoping Review"

_cancers, 2023, doi:10.3390/cancers15184550_

Round 1
Reviewer 1 Report
In this review manuscript, Omer et al nicely summarized the clinical outcomes and limitations of Bispecific T-cell engagers (BiTEs) and bispecific antibodies (BiAbs) in hematological malignancies, focusing on multiple myeloma, non-Hodgkin lymphoma, acute myelogenous leukemia, and acute lymphoblastic leukemia. The authors summarized the resistance mechanisms such as target antigen loss and immunosuppression from TME and provided strategies to overcome the resistance. The manuscript was well written, and the figures were well-designed.
Additional comments:
1. From line 395 to 400, the authors mentioned high neurological events were associated with blinatumomab, AFM11 and duvortuxizumab. Can the authors provide more information or explain why Tnb-486 had a lower incidence of ICANS compared to other CD3XCD19 BiAb?
2. From line 423 to 424, can the authors list ORR of PV alone, glofitabmab alone and mosunetuzumab alone?
3. From line 475 to 477, can the authors explain more why CD33XCD3 BiAbs/BiTEs have been limited? Is it due to the expression of CD33 on hematopoietic stem cells? Since CD123 is also expressed on hematopoietic stem cells. Will CD123XCD3 BiAbs/BiTEs have the similar limitations as CD33XCD3 BiAbs/BiTEs have?
4. The authors shall format the tables better so that the number under N will not be in two lines such as 165 in Table 1.
Author Response
We would like to thank the reviewer for their thorough evaluation of our manuscript.. All reviewer comments have been addressed in the manuscript with tracked changes and are copied below.
Reviewer 1
We thank you for thoroughly reviewing our manuscript.
- From line 395 to 400, the authors mentioned high neurological events were associated with blinatumomab, AFM11 and duvortuxizumab. Can the authors provide more information or explain why Tnb-486 had a lower incidence of ICANS compared to other CD3XCD19 BiAb?
To address this concern, we have made the following change:
“Mechanistically, the lower incidence of ICANS and CRS associated with Tnb-486 is likely due to its unique anti-CD3 moiety designed to bind CD3 on T cells with low affinity, thereby attenuating the release of pro-inflammatory cytokines [118]. “
- From line 423 to 424, can the authors list ORR of PV alone, glofitabmab alone and mosunetuzumab alone?
The response rates for monotherapy with these BiAbs have now been included to clarify the difference in ORR between monotherapy and combination therapy.
“The antibody-drug conjugate polatuzumab vedotin (PV) targets CD79b, an antigen that is expressed on the majority of malignant B cells in NHL [126]. Two phase Ib/II clinical trials assessing the efficacy and safety of glofitamab and mosunetuzumab in combination with PV have demonstrated promising results with ORR rates of 80% with glofitamab and 72% with mosunetuzumab in RR DLBCL [110,125]. The ORR observed with this combination appear to be superior to those reported with both glofitamab monotherapy (ORR 58.0%) and mosunetuzumab monotherapy (ORR 42.0%) in RR DLBCL.”
- From line 475 to 477, can the authors explain more why CD33XCD3 BiAbs/BiTEs have been limited? Is it due to the expression of CD33 on hematopoietic stem cells? Since CD123 is also expressed on hematopoietic stem cells. Will CD123XCD3 BiAbs/BiTEs have the similar limitations as CD33XCD3 BiAbs/BiTEs have?
We have amended this section and removed the sentence “However, the development of CD33xCD3 BiAb has been limited.” Thus, for the sake of clarity and highlighting the landscape of CD33xCD3 BiAb development we have added the following sentence: “Clinical trials evaluating the efficacy of anti-CD33 bispecific T cell engagers such as AMG 673, AMG-330, and GEM333 have been terminated despite promising preliminary results; however, there are two CD33xCD3 BiAbs in clinical development after completion of initial phase 1 studies (JNJ-67561244 and AMV564).”
- The authors shall format the tables better so that the number under N will not be in two lines such as 165 in Table 1.
We have now re-formatted the table to correct this error.

Reviewer 2 Report
The manuscript by Omer et al. provides a precise and thorough review of bispecific T cell engagers (BiTEs) and bispecific antibodies (BiAbs), evaluating the therapeutic potential of most such reagents now approved for in vivo use in a variety of mainly haematological diseases. The authors not only assess the advancements in treating different pathologies, but also delve into the spectrum of adverse events (AE) most frequently associated with these reagents.
The majority of BiTEs and BiAbs show far better results than other forms of conventional therapy, albeit with variations contingent on specific diseases and lineages. In addition to addressing the prevalent AEs, the authors are also aware of and acknowledge the existence of resistance. Cytokine release syndrome (CRS) is a common complication demanding immediate attention. Other significant adverse events are haematological and neuro-toxicities, which are more manageable in clinics.
One particularly interesting section of the manuscript deals with the search for new tumor targets, now still traditionally link to the historical CD38 generation. Numerous promising molecules are currently under intense investigation. Another compelling facet of the manuscript pertains to elucidating potential mechanisms of resistance to these innovative reagents. Notably, the decline in surface target molecule following prolonged therapy, often accompanied by the selection of cell clones not expressing the molecule under analysis, emerges as a plausible explanation. Another possibility is the induction of exhaustion of the effector cells. Fig. 1 provides a good visual of the events taking place in multiple myeloma (MM) therapy using anti-BCMA and -GPRC5D BiTEs. The figure also proposes possible strategies for overcoming resistance in MM.
Fig. 2 presents a schematic view of the mechanisms of action of CD19xCD3, the first reagent approved in the field. Of general interest is that the figure contains a list of potential applications of the antibody. Furthermore, the mechanisms of resistance to the antibody are commented, with special attention to the modulation of the tumor microenvironment (TME). Expansion of this aspect would be helpful.
The manuscript also analyses scenarios outside MM and B-ALL, evaluating new targets across the effectors and a panel of surface molecules expressed by Non-Hodgkin lymphoma cells. The strengths and limitations of these diverse choices are summarized in Fig. 3.
In Fig. 4 the authors provide an outline of the most significant toxicities induced by BiTEs in hematological malignancies. This chart will likely be found useful for teaching purposes as well as in practical clinical use.
The manuscript draws multiple conclusions, advocating for the use of these reagents in early phases of disease management. The exploration of combinations comprising both mono- and bi-specific antibodies emerges as an intriguing avenue of research. Such combinations may improve in function either by adopting multiple specificities or by modulating the action of the Fc of the antibody.
Author Response
We thank you for thoroughly reviewing our manuscript. To address your comment on expanding the section on the immunosuppressive TME as a mechanism of resistance to blinatumomab, we have made the following edits to lines 332-348:
“Lastly, the immunosuppressive TME in ALL may promote resistance to blinatumomab [86]. A higher burden of T-regs has been associated with resistance to blinatumomab, whereas a greater presence of CD8+ effector and memory T-cells and of CD3+ T cells is associated with a better response to treatment [13,87]. B-ALL patients who do not respond to blinatumomab exhibit T-cell deficiency in the TME and higher levels of immune checkpoint molecules such as PD-1, TIM-3, and TIGIT compared to responders [87,88]. In agreement with these findings, a recent phase 2 clinical trial on patients with chronic lymphocytic leukemia (CLL) and Richter’s transformation to diffuse large B-cell lymphoma (DLBCL) showed that complete responders to blinatumomab expressed the lowest levels of PD-1, TIM-3, and TIGIT [89]. T-cell exhaustion may be related to exposure to multiple lines of cancer therapy prior to blinatumomab as these agents are not typically used as first-line treatments or from continuous exposure to blinatumomab, with the persistent T-cell stimulation causing subsequent exhaustion [16,17]. Accordingly, strategies to reprogram the immunosuppressive TME include treatment-free intervals, which can reduce T-cell exhaustion, and the use of ICIs such as nivolumab and pembrolizumab [16]. Results from early stage-clinical trials demonstrate that combining ICIs with blinatumomab is safe; however, efficacy results are still awaited [90,91].”
